# Pattern Detection Model Using a Deep Learning Algorithm for Power Data Analysis in Abnormal Conditions

**Jeong-Hee Lee [1] , Jongseok Kang [2,\*], We Shim [2], Hyun-Sang Chung [2] and Tae-Eung Sung [1,\*]**

[1] Department of Computer Science, Graduate School, Yonsei University, Wonju 26493, Korea; smartbio@naver.com
[2] Division of Data Analysis, Korea Institute of Science and Technology Information (KISTI), Busan 48058, Korea; sw@kisti.re.kr (W.S.); hschung@kisti.re.kr (H.-S.C.)
\* Correspondence: kangjs@kisti.re.kr (J.K.); tesung@yonsei.ac.kr (T.-E.S.)

**Abstract:** Building a pattern detection model using a deep learning algorithm for data collected from manufacturing sites is an effective way for to perform decision-making and assess business feasibility for enterprises, by providing the results and implications of the patterns analysis of big data occurring at manufacturing sites. To identify the threshold of the abnormal pattern requires collaboration between data analysts and manufacturing process experts, but it is practically difficult and time-consuming. This paper suggests how to derive the threshold setting of the abnormal pattern without manual labelling by process experts, and offers a prediction algorithm to predict the potentials of future failures in advance by using the hybrid Convolutional Neural Networks (CNN)–Long Short-Term Memory (LSTM) algorithm, and the Fast Fourier Transform (FFT) technique. We found that it is easier to detect abnormal patterns that cannot be found in the existing time domain after preprocessing the data set through FFT. Our study shows that both train loss and test loss were well developed, with near zero convergence with the lowest loss rate compared to existing models such as LSTM. Our proposition for the model and our method of preprocessing the data greatly helps in understanding the abnormal pattern of unlabeled big data produced at the manufacturing site, and can be a strong foundation for detecting the threshold of the abnormal pattern of big data occurring at manufacturing sites.

**Keywords:** manufacturing big-data; fault detection; abnormal pattern; labelling; neural network; hybrid model; CNN-LSTM; pattern search model; Fast Fourier Transform; frequency domain

## 1. Introduction

Recently, the research into big data in scientific and technological fields has been rapidly increasing, and there is an increasing trend of attempts to improve the competence and competitiveness of companies, and discover new business items, using manufacturing big data information from the manufacturing industry.

Data obtained through sensors at the manufacturing site play an important role in the successful operation of the manufacturing plant. The manufacturing company can get an overall quality improvement and cost reduction through defect tracking, and predictive function improvement using machine learning. Data obtained through sensors at the manufacturing site play an important role in the successful operation of the manufacturing plant, and the manufacturing company can get an overall quality improvement and cost reduction through defect tracking and predictive function improvement using machine learning. Big data in the manufacturing arena is expected to be helpful for enhancing process efficiency and the prediction of industrial accident symptoms through machine learning.

Big data in manufacturing environments need to be analyzed in combination with a variety of data in terms of size or format, in order to gain new insights that are not previously held, and then extended, as a means of integrating a series of tasks, people and business processes that increase productivity [1].

According to the 2011 McKinsey Global Institute data, the manufacturing industry accumulates data notably larger than other industries, and as of 2009, US companies had 966 petabytes of data regarding discrete manufacturing and 694 petabytes regarding process manufacturing industries, such as chemical processing [2].

Equipment in the manufacturing industry is composed of automation equipment, and log data in a numerical or text format is stored from the equipment or sensors installed nearby; the operation of manufacturing facility is then managed based on these data. The operational management methodologies for the maintenance of manufacturing facilities include (1) the reactive method, (2) the predictive method, (3) the proactive failure method, and (4) the self-maintenance method [3].

Reactive maintenance refers to maintenance performed after a failure has occurred, and predictive maintenance is the method of performing maintenance to avoid failures according to conditions based on time. In addition, proactive maintenance is a method of eliminating the cause of the failure in advance, and self-maintenance is a method of self-maintaining, from failure detection to action [3].

In manufacturing sites, a 24 h continuous, seamless process is in progress, and sometimes situations occur in which the process environment changes unexpectedly, and the manufacturing process repeats the production line stops and resumptions due to equipment maintenance or other reasons, so in the database data may accumulate that exceeds the normal range, or that results in significant deviation due to malfunctioning sensors (though still in the normal range). For this reason, it is difficult to identify patterns of abnormal data in the manufacturing process.

The labelling of anomalous data is exceedingly difficult without domain knowledge of the manufacturing process, and collaboration with manufacturing process experts enables data analysis for deep learning after the labeling of abnormal data patterns, but it is practically difficult and time-consuming for data analysis professionals to obtain help from manufacturing process experts at actual manufacturing sites. Accordingly, there is a need for an easy and convenient methodology of identifying patterns of abnormal data from the manufacturing process.

This study focuses on deriving the threshold setting of the abnormal pattern without manual labelling of big data occurring at manufacturing sites, and building a manufacturing failure prediction algorithm to recognize unexpected failures and accidents in advance without the domain knowledge of the manufacturing process.

The data preprocessing method which involves the Fast Fourier Transform (FFT) technique enables us to detect the threshold of the abnormal pattern that cannot be found in the existing time domain. Our proposed hybrid Convolutional Neural Networks (CNN)–Long Short-Term Memory (LSTM) algorithm after FFT preprocessing shows relative superiority over other widely spread algorithms, such as the typical Deep Neural Networks (DNN), Recurrent Neural Network (RNN) and Long Short-Term Memory (LSTM), in terms of both loss and the accuracy of either the prediction or classification model.

The remainder of the paper is organized as follows. The Section 2 reviews current studies of big data-based machine learning, and the Section 3 describes the basic algorithms and techniques used in this study in detail. The Section 4 explains the method proposed in our study, and lastly, the Section 5 contains the result of this study, followed by the final section presenting the discussion and conclusion.

## 2. Related Work

### 2.1. Application of Machine Learning Algorithms for the Detection of Fault and Abnormal Patterns

S.I. Na and H.J. Kim proposed an anomaly detection system based on big data collected from solar sensor data, in which the machine learning model of Support Vector Machine (SVM) is applied to

time-series data for voltage, vibration, temperature, humidity and illumination. The pattern detection system proposed herein applies a data preprocessing model that considers the characteristics of the sensor so as to accurately and reliably detect abnormal faults or states in the sensor data. Because the sensor data collected has a time-series structure, an SVM was refined to detect the anomalies [4].

In the study by Lee et al., the Long Short-Term Memory (LSTM) algorithm was used to diagnose the feasible faults of any type of digital sensors by analyzing the rising time (RT) and falling time (FT) of digital sensors. The study proposed a fault diagnosis model that is approximately 50% better than the rule-based fault diagnosis, and also proposed a model for analyzing the signal through LSTM to diagnose the fault type of the sensor with the rising time and falling time. This study ultimately suggested an LSTM model that represents the two output nodes of failure and normal condition, and then utilized the Softmax activation function and the Cross Entropy function to derive the output value. Meanwhile, optimization was performed using the Adaptive Moment Estimation (Adam) optimizer [5].

Google's Convolutional, Long Short-Term Memory and Fully Connected Deep Neural Networks (CLDNN) model, combining Convolutional Neural Networks (CNN) and Long Short-Term Memory (LSTM), showed a 4–6% relative improvement in the Word Error Rate (WER) over an LSTM. CNNs perform well in reducing frequency variations, whereas LSTMs are good at temporal modeling [6]. The CLDNN model is one of the models that can be applied to manufacturing data consisting of time-series data.

Conventional fault diagnosis and classification methods first extract features from the raw process data. The Vanilla LSTM model by H. Zhao et al., based on Batch Normalization (BN), directly classified raw process data without specific feature extraction or classifier design in the fault diagnosis of chemical process data, minimized the loss function effectively, and then showed a promising fault diagnosis performance [7].

Rotating machinery effectively operates under tough conditions and environments, and so it has a higher faulty rate compared with the other parts in a mechanical system. For rotating machinery, the actual measurement signal is very noisy in the time domain, and it is difficult to disclose the inherent nature of the signal. The Long Short-Term Memory Recurrent Neural Network (LSTM RNN) model by Rui Yang et al. used Fast Fourier Transform (FFT) to reduce the computation burden, and the data-driven model showed an effectiveness in detecting the fault and classifying the corresponding fault types superior to conventional mathematical models [8].

The problem of power demand forecasting for the effective planning and operation of smart grid, renewable energy and electricity market bidding systems is an ongoing challenge. Power demand forecasting problems remain one of the challenging issues, since existing methods are not sufficiently practical to be widely deployed due to their limited accuracy. The (c, l)-CNN + LSTM hybrid model by M. Kim et al. demonstrates better performance than previous neural network models, such as the Sequence to Sequence (S2S), Long Short-Term Memory (LSTM) and Autoregressive Integrated Moving Average (ARIMA), with a Mean Absolute Percentage Error (MAPE) of 0.91% in power demand forecasting. In this study, CNN and LSTM are placed horizontally, which combines the outputs of both networks at the merge layer after feature extraction by CNN and LSTM [9].

The real-time monitoring and fault diagnosis of bearings are of great significance in terms of improving production safety, preventing major accidents, and reducing production costs. A study by Z. Zhuang et al. proposed a Stacked Residual Dilated Convolutional Neural Network (SRDCNN)-based Intelligent Fault Diagnosis Method, and reported that the proposed SRDCNN model had superior denoising ability and a better workload adaptability [10].

In general, the bearing's weak fault feature exhibits a nonlinear and non-stationary nature, which is hard to extract given the situation of existing strong background noise and interference components. The study by M. Ge et al. showed that the Local Robust Principal Component Analysis (LRPCA) can decompose the signal trajectory matrix into multiple low-rank matrices, and suppress the noise. Multi-Scale Permutation Entropy (MSPE) was used to identify the low-rank matrices corresponding to

the bearing's fault feature. Mao Ge et al.'s model using LRPCA and MSPE can effectively detect and locate the bearing faults accurately [11].

Principal component analysis (PCA) is widely used in fault diagnosis, and Z. Wang et al. proposed the data preprocessing method, based on the Gap metric in the Riemann, space to improve the performance of PCA in fault diagnosis. The proposed method can detect the source of the fault more accurately, and reduce the rate of misdiagnosis and the rate of omissive judgement [12].

Tool fault diagnosis in numerical control (NC) machines plays a significant role in ensuring manufacturing quality, but current methods of tool fault diagnosis lack accuracy. C. Gao et al. proposed the fault diagnosis method based on the Stationary Subspace Analysis (SSA) and Least Squares Support Vector machine (LS-SVM). The proposed method has better diagnosis accuracy than the three previous methods, based on LS-SVM alone, principal component analysis and LS-SVM, or on SSA and Linear discriminant analysis [13].

H. Qin et al. offered an end-to-end one-dimension convolution neural network (1D-CNN) model for fault diagnosis of rolling bearings using vibration signals, which showed improved accuracy and proved the effectiveness of 1D-CNN-based fault diagnosis [14].

### 2.2. Application of Machine Learning Algorithms for Solving the Data Imbalance Problem

Next, the imbalance problem is also one of the top 10 most challenging problems in data mining, and class imbalance problems have been reported to occur in a wide variety of real-world domains [11]. In recent years, the manufacturing industry has recognized this class imbalance as a major barrier to the advancement of high-performance fault detection models [15].

The Deep Convolutional Neural Networks (DCNN) approach is currently producing a high resolution for a variety of problems, but time-series data recognition requires continuous label prediction, rather than a single label, unlike normal data recognition. In the study by B. Shi et al., the Convolutional Recurrent Neural Network (CRNN) model, combining DCNN and recurrent neural networks (RNN) in a cascaded manner, showed that without preprocessing steps, time-series labels that do not require detailed annotations can be learned directly, and the CRNN model is not constrained by the length of the time-series data [16]. Most manufacturing-related sensing data consists of time-series data. The CRNN model can be applied to manufacturing big data consisting of time-series data.

When learning from an imbalanced data set, models often create data that are more accurate only for classes in a high proportion of training data. To cope with this issue, Synthetic Minority Over-sampling Technique (SMOTE) algorithms are widely used to create additional low-proportional classes of data by giving random characteristics to existing data in unbalanced data sets, increasing the number of sparse classes. The study by J.-H. Seo used SMOTE techniques to improve the efficiency of classification methods, such as the k-NN (k-Nearest Neighbor) Test, the SVM Test, and the Decision Tree Test [17].

N.V. Chawla et al. proposed the SMOTEBoost algorithm that combines the Synthetic Minority Oversampling Technique (SMOTE) and the standard boosting procedure. SMOTEBoost can create synthetic examples from the rare or minority class. SMOTEBoost, applied to several highly and moderately imbalanced data sets, showed improvements in prediction performance for the minority class, and overall improved F-values [18].

Y. Sun proposed a cost-sensitive boosting algorithm, introducing cost items into the learning framework of AdaBoost (Adaptive Boosting), a kind of ensemble-based classifier, and Y. Sun's study showed that the cost-sensitive boosting algorithm can improve predictive performance for minority classes [19].

Y. Lim proposed a Value-assigning Neural Nets (VAN) model for improving the performance of classification models via a cost-assigning process of data. In the VAN model, the evaluation module based on the auto-encoder model gave an appropriate value by considering the characteristics of the data, and the classification module considered the imbalance based on the value of the major data and improved the classification performance. The VAN model led to an improved classification

performance compared to existing models, such as Adaboost (Adaptive boosting) and CHI-BD (A fuzzy rule-based classification system for Big Data classification problems), for the unbalanced data [16].

In the case of corporate bankruptcies prediction problems (data imbalance problems), it is common that the class representing the normal case unilaterally has a higher rate than the case of company defaults. Ahn's study combining the Random Over Sampling Examples (ROSE) method with Support Vector Machine (SVM) demonstrated that the ROSE method can contribute to the improvement of the prediction accuracy of SVM in the Bankruptcy Prediction Model [20].

Lee's hierarchical model, hierarchically combining the ensemble of under-sampling support vector machines, demonstrated that the accuracy of the minority class is maintained at a certain level, and the accuracy of the majority class is improved in the test classifying the death accidents from actual individual traffic accident data [21].

A study by Shin, applying SMOTE in the potential space to match the ratio of the minority category to the majority category, and using a machine learning technique called Adversarial Auto-Encoder (AAE), contributed to improving the performance of unbalanced data classification by performing over-sampling in the latent variable space where feature extraction performed strongly [22].

From the literature review above, we determined that we often feel the necessity to find an effective means of setting the threshold of the abnormal pattern in data which are not labeled for fault detection tasks in manufacturing sites. Most of the previous studies utilized the labeled data for the training to build software learning models for fault detection patterns. The algorithm most frequently used in previous studies is LSTM, and the machine-learning algorithms that have been tried recently are PCA, SVM and VAN. Out of those, SVM and VAN have recently been newly tested to solve the data imbalance problem. In this paper, we tried to address these issues by the proposed hybrid CNN-LSTM algorithm and the Fast Fourier Transform (FFT) technique, where we perform the preprocessing of the data set through FFT, which enables us to detect anomalies that cannot be found in the existing time domain.

## 3. Background

As we have investigated in the related works of the previous section, there exist various application cases for typical machine learning algorithms. We hereby focus on the manufacturing industry's big data analytics, in the situation wherein a company's business loss becomes fatal because the repair cost of the semiconductor manufacturing equipment's breakdown is too huge, and takes 3 to 6 months at least. This section presents a short introduction of the basic algorithms, such as CNN, RNN and the LSTM algorithm, and techniques for data preprocessing such as Fast Fourier Transform (FFT), to understand the modified algorithms and techniques used in this study.

### 3.1. CNN Algorithm

CNN is a universally used algorithm for two-dimensional image classification, and is structured by connecting the characteristic extraction neural network and the classification neural network in series, as shown in Figure 1. CNN uses three types of layers: the convolutional layer, the pooling layer and the fully-connected (FC) layer. Convolutional layers that handle the convergence serve to explore the same characteristics at various locations in the input image, and the pooling layer is used to reduce the size of the output data (Activation Map), or to emphasize specific data by receiving output data from the convergence layer as input [23].

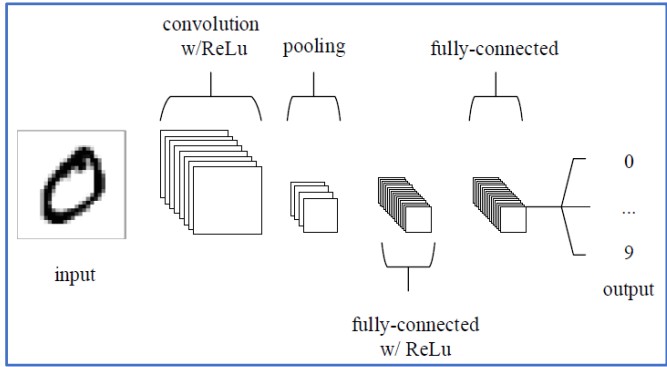

**Figure 1.** A simple CNN (Convolutional Neural Network) architecture.

ConvNet consists of a learning weight and a bias. Each neuron receives input, performs an internal operation (dot product), and then performs a non-linear operation according to its selection. The entire network, like a general neural network, has a single score function that is distinguishable, and a loss function such as Softmax in the last layer [24].

One of the advantages of CNN is that it is able to learn while maintaining spatial information in images, and that it is immutable under conditions of noise and distortion, while the disadvantages of CNN are that it uses a large number of parameters that occupy more memory capacity than a typical multi-layer theory, and that the convergence process in terms of execution time requires a lot of calculations [25].

### 3.2. RNN and LSTM Algorithm

The RNN algorithm is a type of artificial neural network specialized in repetitive and sequential data learning, which uses an internal circulation structure to reflect past learning into current learning through weight, as shown in Figure 2 [26].

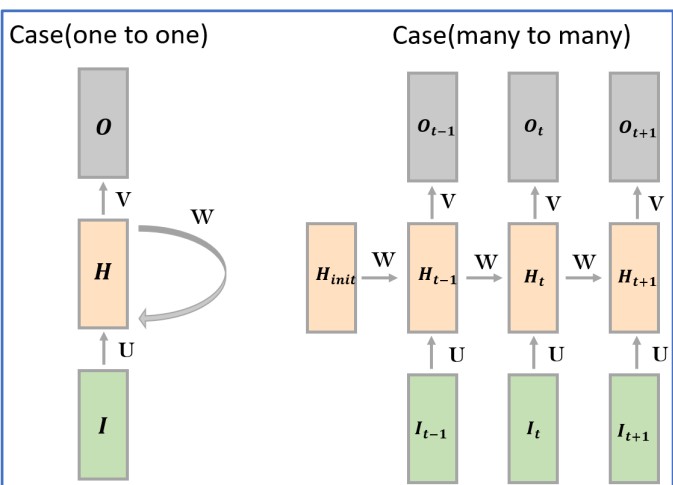

**Figure 2.** RNN (Recurrent Neural Network) algorithm.

The disadvantage of RNN is that there is a long-term dependency problem, in which the value of the weight that started first gradually decreases as learning progresses. The algorithm that complements this is the Long Short-Term Memory Network (LSTM) algorithm introduced by Hochreiter and Schmidhuber [27,28].

LSTM is one of the algorithms belonging to RNN classification using the memory block, shown in Figure 3, and has since been developed further in several studies. The advantage of

LSTM is that each memory can be controlled, and the result can be controlled, but there are also disadvantages in that the memory can be overwritten and the operation speed is slow [26].

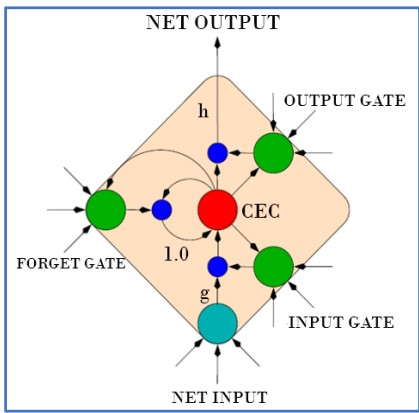

**Figure 3.** LSTM (Long Short-Term Memory Network) memory block.

*3.3. The Fast Fourier Transform (FFT) Technique*

Fast Fourier Transform (FFT) is one of the most useful tools for, and is widely used in, signal processing, as shown in Figure 4. The FFT is not a new transform; it is just a fast algorithm to compute Discrete Fourier Transform (DFT). DFT means converting a discrete signal in the time domain into a discrete signal in the frequency domain. The FFT is based on a halving trick, that is, a trick to compute the DFT of a length N time-series using the DFT of two sub-series, of length N/2 [29]. FFT is a technique that reduces the amount of computation via a reducing of the iterative computation of a DFT, by dividing the calculations each by half.

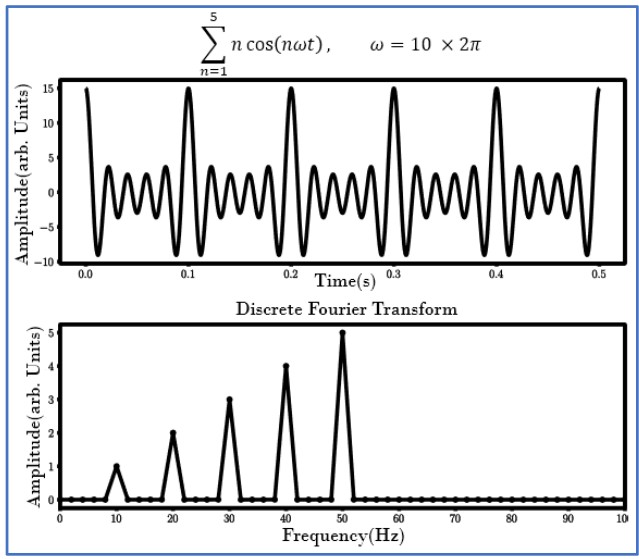

**Figure 4.** An example of Fast Fourier Transform.

Fourier analysis converts a signal from its original domain (often time or space) to a representation in the frequency domain, and vice versa. Fast Fourier Transforms are widely used for applications in engineering, music, science and mathematics. The FFT is used in digital recording, sampling, additive synthesis and pitch correction software [30].

## 4. Proposed Method

This section offers a detailed discussion of the method proposed in this paper, which includes data collection, the applied techniques for data preprocessing [such as Fast Fourier Transform (FFT)] and frequency domain analysis, and the algorithm proposed ultimately as the CNN-LSTM algorithm. All of these are significant in that the abnormal pattern can be reliably detected through the integration of both FFT-frequency domain analysis and CNN-LSTM, which has not been tried yet.

The test platform is explained as follows. First, we measured and collected three types of data, i.e., electric power, temperature and vibration data, in a second unit for 56 days, then performed preprocessing for appropriate labeling, which corresponds to either the normal operation of a machine or the malfunctions due to external conditions. After taking FFT and conducting the relabeling, we apply two representative algorithms of a typical LSTM and a hybrid CNN-LSTM (proposed), under the framework of PyTorch and a 3-GPUs-enabled deep-learning server, while flexibly adjusting the hyper parameters for performance evaluation with regards to accuracy and loss.

### 4.1. Data Collection and Preprocessing

The electric power data of this study is extracted, on a per second basis, from the sensor of unit 29 in the factory of JEONGMIN Electric Co. Ltd., located in Changwon, Gyeongsangnam-do, South Korea, from 13 June 2019 to 19 September 2019. Temperature and vibration data were also collected from other sensors, but as a result of the visual observation of the data set in unit 29, only electric power data were used in this study, except for temperature and vibration variables, which could be changed regardless of malfunction due to external conditions. The temperature measured from the sensor attached to the equipment continued increasing after the initial operating time, and it also displayed a pattern reaching a peak at noon. It was observed that the change in temperature due to the generation of heat with the elapse of the operating time of the machine itself was much larger than the change in the external temperature, and we recognized that it had a considerably greater influence on the temperature measured from the sensor. During the initial operation of the production equipment, it was found that the vibration increased temporarily due to an increase in the operating speed. Due to the existence of differences in peak power time, peak vibration time and peak temperature time in the data, it is difficult to judge changes due to abnormal data, and it is difficult to derive abnormal patterns, so the research was conducted using only power variables, with the exception of temperature and vibration variables. In situations where the presence of different patterns of variables and the malfunctioning situation are not labeled, the use of various variables is judged to result in a poor analytical performance.

Weekend and holiday power data were excluded so as to avoid outliers, since usually the factory does not work on weekends and holidays. We used the power data from 56 days for actual analysis. According to a survey of the data distribution of the used data, cases where power consumption is less than 10,000 kWh constituted about 1% of the total. The descriptive statistics for the data used is shown in Table 1.

**Table 1.** Descriptive Statistics for Electric Power Data (Unit: kWh).

| Data Source | Average | Q1 | Median | Q3 |
|---|---|---|---|---|
| Unit 29 sensor | 39,574 | 18,354 | 36,709 | 55,063 |

As mentioned before in Figure 4, FFT often helps with frequency component-characterized understanding by displaying the frequency components much more clearly, even when the time domain signal is difficult to recognize or it is hard to discern periodic or cyclo-periodic features. The experiments without FFT are performed over the entire data set, which is observed per-second and measured for the last 56 days. Including specific intervals where power the data abruptly increases, we evenly take the time domain signal subsets and their corresponding labeling. Then, we apply two representative algorithms of LSTM and the hybrid CNN-LSTM, which are expected to provide good

performances, in contrast with DNN, which does not reflect the memory characteristics of time-series data and has considerable computational complexity as the number of layers increases, and RNN has the disadvantage of the long-term dependency problem, i.e., gradually decreasing weights as learning progresses.

Therefore, we came up with the novel idea that after extracting the frequency domain characteristics of the abnormal pattern via FFT, and applying the Nyquist–Shannon sampling theorem, we could regroup the second-based time domain data sets into 15 s-based ones, as we observed that the abnormal patterns lasted for about 14 s to 16 s.

Based on a visual review of electric power data, the average duration of abnormal data was observed to be 15 s. For this reason, the data set was prepared by rearranging the data at 15 s intervals, by arranging the power data of 15 time-periods, each generated in one second, into one row.

Power data corresponds to an analog signal that continuously changes in both time and size, and specifically, a discrete signal. Digital signal processing is a process of converting an analog signal into a digital signal, through which it can be used in various fields, such as image processing, audio and audio signal processing, and communication. To analyze power data belonging to discrete data, it is necessary to transform the time domain signal into a frequency domain so that it can be observed. For this transformation, Fast Fourier Transform (FFT) was used.

The overall process including FFT is as shown in Figure 5. We average 15 time-periods' data in a row, with labelling that appropriately discriminates between normal and abnormal patterns in the power data. Then, in order to effectively catch and clearly visualize the distinct time intervals for abnormal labelling, we take the FFT of the preceding time domain's averaged data. In our experiments, we require an interpolation in the frequency domain, where feature observations at specific intervals can be spread into much wider frequency intervals via the assumption of the Nyquist–Shannon sampling theorem. The Nyquist–Shannon sampling theorem implies that a bandlimited continuous-time signal can be sampled and perfectly reconstructed from its samples if the waveform is sampled over two periods that are as fast as its highest frequency component. For example, if we consider the (0, 500 Hz) frequency domain signal, we need to take the sampling frequency of 1000 Hz or larger. Then, we extend the analysis to (0, 3970 Hz) for much clearer visualization in terms of interpolation, if necessary. From the interpolated frequency domain data, the corresponding time domain data are reconstructed, and the relabeling is performed.

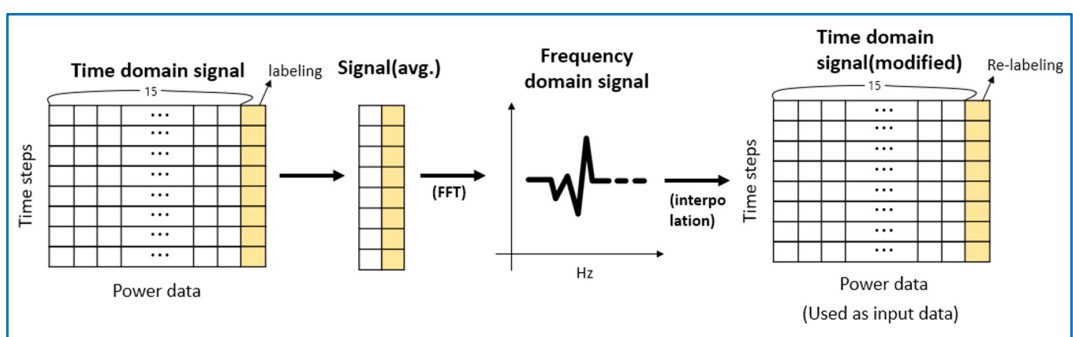

**Figure 5.** Process of data pre-processing.

## 4.2. CNN-LSTM Algorithm

The artificial neural network algorithms mainly used for manufacturing sensing data analysis include CNN and LSTM, and CNN, which is an effective model for the feature extraction of data, and is one of the deep learning techniques that builds up multiple layers of artificial neural networks. CNN offers an excellent performance in image classification through feature extraction, and recently, as data volume has increased, it has also been used for feature extraction in general time-series data. The proposed CNN-LSTM algorithm is as shown in Figure 6.

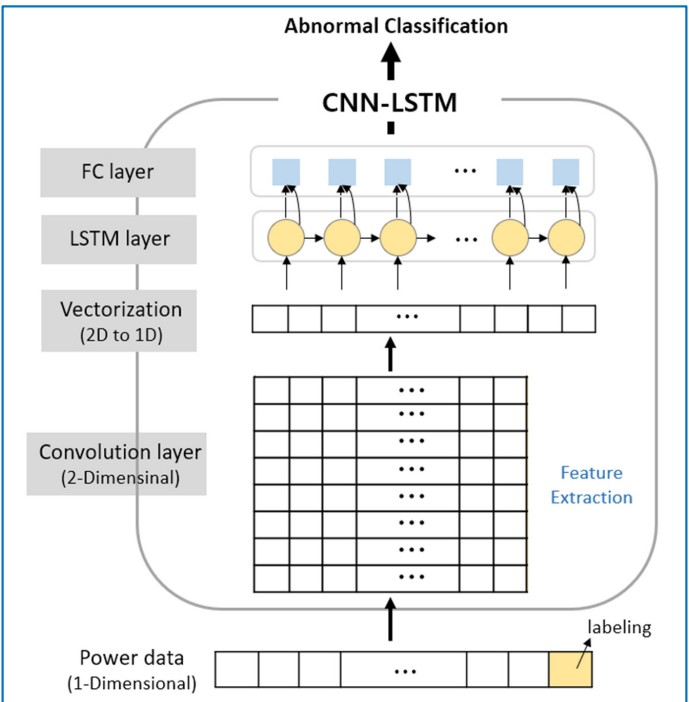

**Figure 6.** Proposed CNN-LSTM algorithm.

The proposed model is designed to maximize predictive performance by introducing a hybrid neural network algorithm that combines CNN, with the advantage of character extraction for time-series data, and LSTM, with the advantage of long-term dependence on time-series data.

The CNN-LSTM model consists of the convolutional layer, the LSTM layer, and the fully-connected layer. Each layer can adjust hyper parameters. Adjusting these parameters can affect the performance depending on the characteristics of the training data. Our experiments measured optimal performance by adjusting parameters. The first convolutional layer, wherein the kernel size is $5 \times 1$ and the number of filters is 20, received $15 \times 1$ data as input. It is difficult to extract stable features if the kernel size cannot cover a periodic time cycle in the data. Therefore, a $5 \times 1$ kernel was used to minimize information loss and extract local features. In the second convolutional layer, the 50 filters were applied to the output of the first layer in order to extract certain features. These extracted results help to better reflect the characteristics of the data, while reducing the number of parameters used in the following LSTM layer. Before applying it to the LSTM algorithm, the process of making two-dimensional data into one-dimensional data, determining the result of the convolutional layer is necessary. The LSTM network had one hidden layer with 10 units. Finally, the output from the LSTM layer is fed into the FCN layer which consists of 10 units that distinctively classify normal and abnormal conditions.

In the proposed model, CNN extracts appropriate features through time from time-series data for power sensing data, and is responsible for the classification of malfunction cases. LSTM was used for the purpose of minimizing information loss for the time-series data of power sensors.

For comparison with the prediction results when using the existing neural network algorithm, we trained the time-series data of electric power using the LSTM model and compared them with the prediction results from our proposed CNN-LSTM model.

## 5. Results

We tried to train the preprocessed data set by the LSTM algorithm at first, but the learning was not achieved appropriately because there was no tendency to decrease loss. After that, we moved towards training the data set by our proposed CNN-LSTM algorithm, but a lack of convergence in loss performance was observed, even though the accuracy of the learning was improved highly.

To cope with a lack of convergence in loss, we converted the data set from the time domain to the frequency domain using the FFT technique. Then, to find frequency domain segments that better detect abnormal patterns in the data set, we performed FFT on various frequencies using the 15-s averaged data. We could ultimately get the best results with the lowest loss rate with our proposed CNN-LSTM algorithm, after taking FFT at 1000 Hz.

### 5.1. Training by LSTM Algorithm

Cross Entropy was used as the loss function, the SGD (Stochastic Gradient Descent) optimizer was used as the optimizer, and the data set was trained using the LSTM algorithm in 200 epochs. As a result, as the number of epochs increased, the loss value decreased. Both the training data set and the test data set had a slightly lower loss value based on the Epoch 100, but we judged that the learning was not achieved because the form was shown to still be vibrating, as seen in Figure 7.

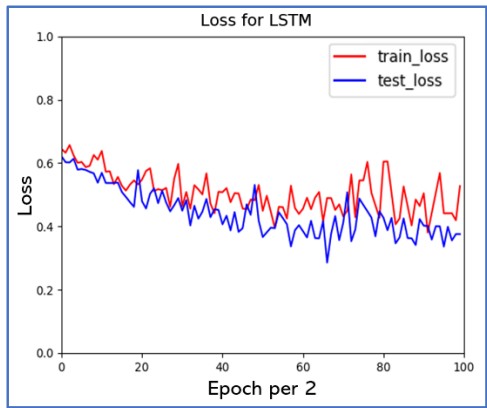

**Figure 7.** Train loss by LSTM algorithm.

As shown in Figure 8, the accuracy of the learning results using the LSTM algorithm was close to about 90%, which was not bad, but this could not guarantee that it had been learnt because there was no tendency to decrease loss.

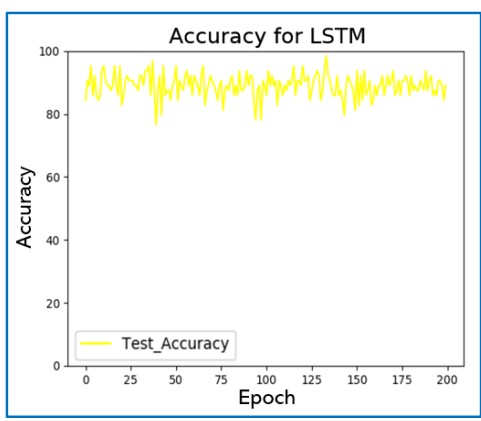

**Figure 8.** Train accuracy by LSTM algorithm.

### 5.2. Training by CNN-LSTM Algorithm

Cross Entropy was used as the loss function, the SGD optimizer was used as the optimizer, and the data set was trained using the CNN-LSTM algorithm in 200 epochs, as shown in Figure 9. The average loss rate was lower when using CNN-LSTM than when using LSTM alone, but it was still vibrating, making it necessary to apply a new method.

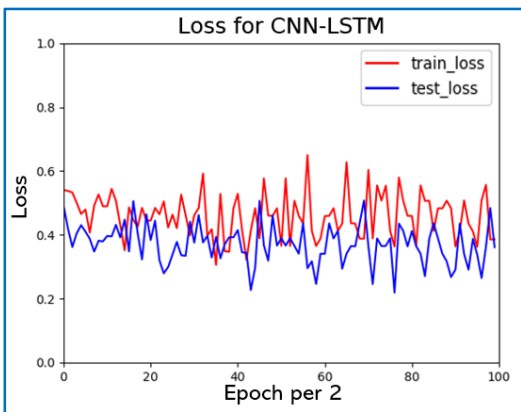

**Figure 9.** Train loss by CNN-LSTM algorithm.

The accuracy of the learning results using the CNN-LSTM algorithm were high, over 90%, as shown in Figure 10, but a lack of convergence was still loss evident, indicating that learning was not proceeding.

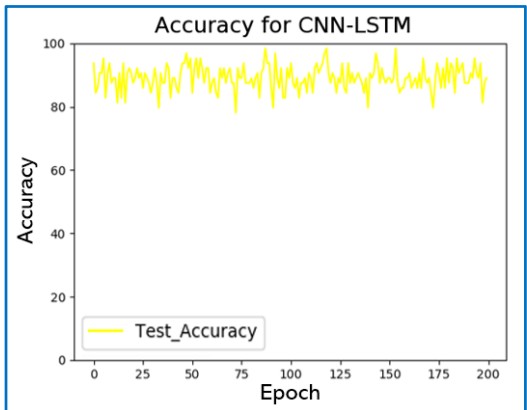

**Figure 10.** Train accuracy by CNN-LSTM algorithm.

*5.3. Training by the CNN-LSTM Algorithm Using the Fast Fourier Transform (FFT) Technique*

Data pre-processing was performed with the time domain data and the frequency domain data by using the FFT technique on the data set, and Figure 11 shows the result of the conversion. Since multiple signals are mixed in the coverage results, additional data preprocessing is required to provide a clear picture of the characteristics of the data.

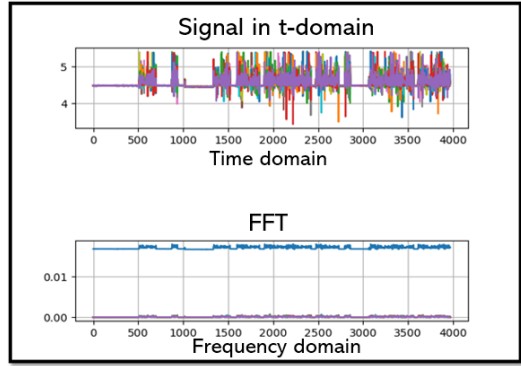

**Figure 11.** Conversion result of data set from time domain to frequency domain.

To clearly understand the characteristics of the data, the average of each row of the data set was obtained, and the data is converted into the form of (3970, 1). The sampling frequency was set to 100 Hz to allow a much clearer distinction between peaks in the frequency domain, as shown in Figure 12.

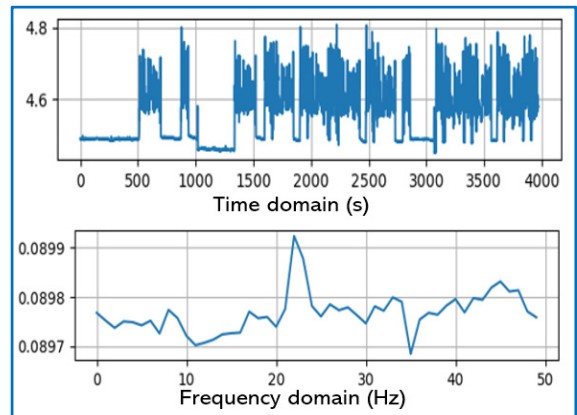

**Figure 12.** Conversion result at 100 Hz frequency using 15-s average data.

It is observed that the overall pattern of loss values becomes much lower when using CNN-LSTM algorithm after FFT using the 15-s average data at a sampling frequency of 100 Hz than when using the CNN-LSTM algorithm without FFT in the training data set, as shown in Figure 13. The graph shows the tendency to be converged in loss over epoch axis and the loss in the test data set did not show much difference from previous experiments.

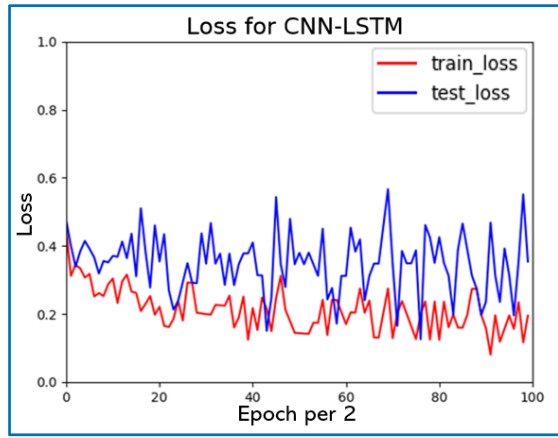

**Figure 13.** Train loss by CNN-LSTM algorithm after FFT at 100 Hz frequency.

In the case of a new data set being pre-processed by applying Digital Signal Processing (DSP), such as FFT, it was confirmed by Train Loss Graph that learning was taking place.

FFT was performed at various frequencies (250 Hz, 500 Hz and 1000 Hz) to find patterns that better detect abnormalities in the data set through the conversion of the sampling frequency, and the results are shown in Figure 14.

The FFT results at various frequencies were able to identify anomalies in similar areas of sampling frequency, as demonstrated in Figure 14. It is judged that the abnormal pattern at 1000 Hz is derived from normal and abnormal patterns in the appropriate range. In the case of frequency domain 1000 Hz, an anomaly was found in three locations in total, and was interpolated from size 0 to size 500, to size 0, then to 3970. The (83–95), (1059–1075) and (1322–1375) parts in the frequency domain where interpolation was performed were assumed to be abnormal. Furthermore, it was assumed that (2595–2648), (2895–2911) and (3875–3887) at the symmetric points, based on Nyquist frequency (1985), were abnormal.

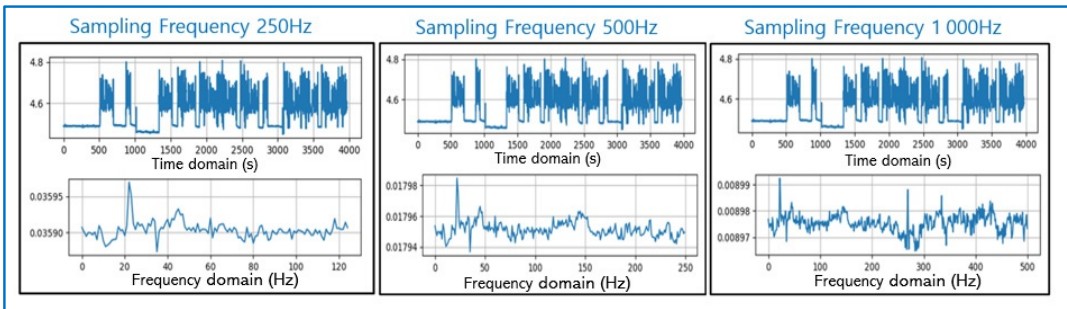

**Figure 14.** Conversion result at 250 Hz, 500 Hz and 1000 Hz.

Based on these assumptions, the study was conducted with new labeling of the existing power data set. We have conducted supplementary experiments to confirm that both the preprocessing method and the proposed model are superior to other deep learning-based models. The following table shows the performance results for each model. As can be seen in Table 2, the proposed model outperforms all others in terms of error indices. Compared to the previous models, the lowest loss rate was confirmed. In addition, we found that both train loss and test loss were well developed with nearly zero convergence, as shown in Figure 15.

**Table 2.** Performance comparison with other deep learning-based models.

| Method | Train Loss | Test Loss |
|---|---|---|
| LSTM | 0.527541 | 0.3756 |
| LSTM * | 0.345669 | 0.3466 |
| Proposed CNN-LSTM | 0.386002 | 0.3614 |
| Proposed CNN-LSTM * | 0.195205 | 0.119716 |

* re-labeling.

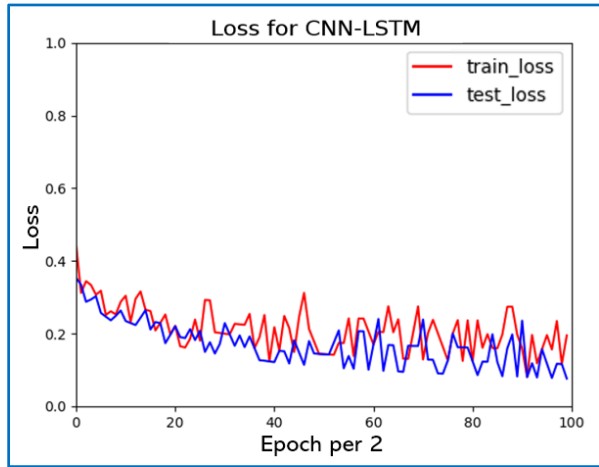

**Figure 15.** Train loss by CNN-LSTM after FFT (Fast Fourier Transform) at 1000 Hz and interpolation.

## 6. Discussion: Open Innovation and Deep Learning

Industry 4.0 is perceived as an innovative approach to the field of manufacturing management, because it contributes to enterprise efficiency and the improvement of competitiveness [31]. Unfortunately, these innovations trigger the growth limits of capitalism [32]. So, from the perspective of open innovation, destructive innovation is not selective, but essential for any firm to survive in market [33]. In line with this, the development of new technologies in the digital age acts as an essential element for maintaining the competitiveness of companies [34]. Among them, artificial intelligence (AI) has been altering industries, as evidenced by companies that have embraced its use to implement

new open innovation business models, and companies that successfully capitalize on AI can create disruptive innovation through their new business models and processes [35]. As an open innovation strategy of the manufacturing industry, "Smart Manufacturing" has attracted considerable attention in the fields of academia and practice, and there is a big gap between policy promotion and production at the site of actual industrial manufacturing [36].

Most organizations use enterprise software, which makes use of rule-based processing to automate business processes including the manufacturing process, and this task-based automation has helped organizations in improving their productivity in a few specific processes. However, this rule-based software cannot self-learn and improve with experience. By analyzing the data, the AI systems can draw conclusions on the machine's condition, and detect irregularities to provide predictive maintenance. A strongly coupled digitized system in the manufacturing industry improves the overall quality and reduces the cost, by improving defect tracking and forecasting abilities [37]. Manufacturing systems engineering will be greatly affected by advances in technology, including the development of cheaper, more ubiquitous sensors, the increase of computational speeds, the ability to hold more data and move it faster, and the development of artificial intelligence in the future [38]. We expect our proposed pattern detection model, using a deep learning algorithm, could be an effective means of decision-making that determines the proper timing of predictive maintenance, with an enhanced pattern tracking ability.

The purpose of this study is to enhance the decision-making and business feasibility of enterprises by providing the results and implications of field-applicable patterns analysis of the big data occurring at manufacturing sites. One of the most important revenue models in the biopharmaceutical industry is a result of big manufacturing capacity [39]. The larger the manufacturing capacity, the more data accumulated, and the more difficult the data analysis.

To utilize data from the collected manufacturing sites for artificial neural networks, the data set types for rules and reasoning corresponding to specific phenomena were identified and verified in various ways.

In the case of power data in which abnormal patterns were not labeled, it was confirmed that detecting and training abnormal patterns through DSP and through FFT offered a more accurate performance than qualitative labeling manually by an experimenter.

Existing studies of randomly generating anomalies to derive data on process anomalies can have a huge negative impact on the sales of small- and medium-sized enterprises, due to their nature in terms of the size of the enterprise. For this reason, based on general process sensing data, the initial data analysis was performed by randomly setting the criteria for the estimated abnormal data, and the search for time-series patterns with regards to the power, temperature and vibration detected during the process was performed for the threshold setting of the abnormal pattern.

## 7. Conclusions

In this research work, our proposed hybrid CNN-LSTM algorithm and the Fast Fourier Transform (FFT) technique present effective ways to set the threshold of the abnormal pattern from data that were not labeled for the fault detection task at manufacturing sites, We also showed that both train loss and test loss were well developed, with near zero convergence with the lowest loss rate, compared to existing models such as LSTM.

Based on the previous research Rui Yang conducted in 2018 [8], the data in the time domain was converted to the frequency domain through FFT, and a new abnormal pattern derivation process was performed. As a result of learning at various frequencies after conversion, it was confirmed that our model can easily find abnormal patterns, and has an excellent learning performance. By converting the data to the frequency domain by FFT, it was possible to detect anomalies that cannot be found in the existing time domain.

In numerical results, we have verified the model's excellence in both loss and accuracy, which determine that the learning performance in the proposed hybrid algorithm, the CNN-LSTM algorithm, is superior to that of widely used existing algorithms, such as LSTM.

This study was performed based on data generated during the manufacturing process over 3 months, and there were limitations in that the target period for detecting patterns was short. It cannot be ruled out that no abnormalities existed in the 3-month period during which the set was recorded.

The limitations of this study are that the study was performed on the data collected in a relatively short period of time, and that only power data, out of various types of data (e.g., vibration, temperature, humidity data, etc.), were analyzed in order to characterize the abnormal conditions of an equipment or a machinery. In future studies, we will move towards conducting an extensive study that increases the reliability of the current research by analyzing the pattern detection model we propose over long-term time-series data, for a year or longer, and all available data, including temperature and pressure data, will be applied in the detecting of anomalous patterns.

If longer-term time-series data were to be used for learning, it would be possible to establish a sophisticated model that can be applied to the industry immediately, because it would be relatively more likely that abnormalities in the process have actually occurred.

The data preprocessing method via the FFT technique, which converts to the time domain and frequency domain, greatly helps to understand the abnormal patterns of unlabeled big data produced at the manufacturing site, and is judged to be an effective way to use deep learning methods with a high learning performance without pre-labeling.

The results of this study are judged to be available as a manufacturing failure prediction algorithm, which alerts of future failures and accidents in advance through past big data analysis in the manufacturing sector. In the case of semiconductor manufacturing and processing equipment, huge fixing costs and over 2–3 months' repair time are required when it is out of order. Recently, the field of predictive maintenance (PdM) has been standing out due to its significance in preventing unexpected stops of machinery in the manufacturing industry. Therefore, we anticipate that our extensive study of long-term manufacturing-related data of various types would contribute to, or develop, PdM, which monitors the performance and condition of equipment during normal operation in order to reduce the likelihood of failures.

Our proposed model and method of preprocessing the data greatly helps in understand the abnormal pattern of unlabeled big data produced at the manufacturing site, and can be a strong foundation for detecting the threshold of the abnormal pattern of big data occurring at manufacturing sites.

**Author Contributions:** Conceptualization, J.-H.L. and T.-E.S.; data curation, J.K. and W.S.; formal analysis, J.K. and W.S.; methodology, J.-H.L., H.-S.C. and T.-E.S.; writing—original draft, J.-H.L.; writing—review and editing, T.-E.S. All authors have read and agreed to the published version of the manuscript.

**Funding:** This research received no external funding.

**Conflicts of Interest:** The authors declare no conflict of interest.

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
