# Peer review of "Pattern Detection Model Using a Deep Learning Algorithm for Power Data Analysis in Abnormal Conditions"

_electronics, doi:10.3390/electronics9071140_

Round 1
Reviewer 1 Report
please see the attached file

Author Response
We would greatly appreciate the referees’ valuable comments and suggestions. We hereby explain to clarify all the points brought up by the reviewers and the responses and are provided in blue color. We hope that the revised version and answers would help the understanding of the reviewers and the readers.
< Reviewer 1 Comment >
1) L31 improving the competitiveness of companies, you have to double check the meaning of the sentence.
Recently, the interest in scientific and technological utilization and research of big data has been rapidly increasing, and its usability has been increasing in various fields such as improving the competitiveness of companies and discovering new businesses through the use of manufacturing big data information. |
=> We appreciate your valuable comment. In order to avoid ambiguity, we modified the expression of the sentence.
Parts revised (in blue color) |
|
Recently, the interest in scientific and technological utilization and research of big data has been rapidly increasing, and its usability has been increasing in various fields such as improving the competitiveness of companies and discovering new businesses through the use of manufacturing big data information. |
Recently, the research of big data in scientific and technological utilization has been rapidly increasing. There is an increasing trend of attempts to improve the competence and competitiveness of companies and discover new business items using big data information in manufacturing industry.
|
2) L 34-37 and other You need to write simple and small sentences as much as possible. I have noticed in different places that the number of word in each sentence sometimes reaches more than 40 words which is unpreferable. Also, there are many points/ ideas in the same sentence and this deteriorate your writing style. The example below is one sentence in the whole paragraph, which is rarely seen in the academic writing style. I suggest to fix all these issue in your writing.
Data obtained through sensors at the manufacturing site plays an important role in the successful operation of the manufacturing plant, overall quality improvement and cost reduction through defect tracking and predictive function improvement, and machine learning, and it is expected to be used to enhance process efficiency and prediction of industrial accident symptoms through machine learning |
|
=> We appreciate your thankful comments. We split the original sentence to each small one and partially modified the expression of the paragraph according to your kind advice.
Parts revised (in blue color) |
|
Data obtained through sensors at the manufacturing site plays an important role in the successful operation of the manufacturing plant, overall quality improvement and cost reduction through defect tracking and predictive function improvement, and machine learning, and it is expected to be used to enhance process efficiency and prediction of industrial accident symptoms through machine learning. |
Data obtained through sensors at the manufacturing site plays an important role in the successful operation of the manufacturing plant. Manufacturing company can get the overall quality improvement and cost reduction through defect tracking and predictive function improvement using machine learning. Big data in manufacturing area is expected to be helpful to enhance process efficiency and prediction of industrial accident symptoms through machine learning. |
3) L55 What does FFT refer to ? First time you mention the abbreviations you have to tell the reader what are these refer to. Similar question could be arise for CNN LSTM RNN DNN abbreviations and other
The data preprocessing method which involves the FFT technique enables us to detect the |
=> We appreciate your valuable suggestion. We appended the description of the abbreviation as follows, because we missed to mention the full description of the abbreviation earlier.
Parts Added (in red color) |
Line 243. DFT means converting a discrete signal in the time domain into a discrete signal in the frequency domain. |
Parts revised (in blue color) |
|
FFT |
Fast Fourier Transform (FFT) |
CNN-LSTM |
Convolutional Neural Networks (CNN)-Long Short-Term Memory (LSTM) |
DNN |
Deep Neural Networks (DNN) |
RNN |
Recurrent Neural Network (RNN) |
LSTM |
Long Short-Term Memory (LSTM) |
DFT |
Discrete Fourier Transform (DFT) |
4) L89, L224 You has to write only SVM / RNN ….. because you earlier defined what is SVM means.
sensor data. Because the sensor data collected has a time series structure, a Support Vector Machine (SVM) algorithm was improved to detect the anomalies
The RNN (Recurrent Neural Network) algorithm is a type of artificial neural network specialized |
=> We appreciate this valuable comment. We deleted the description of the abbreviation like followings because we earlier defined.
Parts revised (in blue color) |
|
Support Vector Machine (SVM) |
SVM |
RNN (Recurrent Neural Network) |
RNN |
5) L91 Please check this expression carefully “referred to as”
In the study by Lee et al., one of the deep learning RNN algorithms, referred to as the Long Short Term Memory (LSTM), was used to diagnose |
=> We appreciate this valuable comment. We modified the expression of the sentence to avoid ambiguity.
Parts revised (in blue color) |
|
In the study by Lee et al., one of the deep learning RNN algorithms, referred to as the Long Short Term Memory (LSTM), was used to diagnose feasible faults of any type of digital sensors by analyzing the rising time (RT) and falling time (FT) of digital sensors. |
In the study by Lee et al., the Long Short Term Memory (LSTM) algorithm was used to diagnose feasible faults of any type of digital sensors by analyzing the rising time (RT) and falling time (FT) of digital sensors. |
6) L223 It seems that there is a mistake in the subtitle, in the context you talking about the RNN; however, the title is LSTM algorithm Please check
3.2. LSTM Algorithm |
=> We appreciate this valuable comment. We modified the mistake in the subtitle.
Parts revised (in blue color) |
|
3.2. LSTM Algorithm |
3.2. RNN and LSTM Algorithm |
7) Figures and tables double check that you referred to the right figure in the manuscript also the location of the figure should be located in the same and relevant section. Further, any figure must be cited in the text. Also, the figures caption should be unified over all the manuscript according to the journal requirements. |
=> We appreciate this valuable comment. We checked the right location of the figure and we modified as in the followings. In addition, we rephrased the right figure in the manuscript to correct the missed figure or misquoted figure.
We unified the figure caption over all the manuscript according to the journal requirement.
1) Font: Palatino Linoty 9, 2) Text alignment : Justify
For Figure 5 & 6, we modified the original caption in bold to one in plain.
Parts revised (in blue color) |
|
CNN (Convolutional Neural Networks) is a universally used algorithm for two-dimensional image classification and is structured by connecting the characteristic extraction neural network and the classification neural network in series. |
CNN (Convolutional Neural Networks) is a universally used algorithm for two-dimensional image classification and is structured by connecting the characteristic extraction neural network and the classification neural network in series, as shown in Figure 1. |
which uses internal circulation structure to reflect past learning into current learning through weight, as shown in Figure 1 [26]. |
which uses internal circulation structure to reflect past learning into current learning through weight, as shown in Figure 2 [26]. |
LSTM is one of the algorithms belonging to the RNN classification using the memory block shown in Figure 2 |
LSTM is one of the algorithms belonging to the RNN classification using the memory block shown in Figure 3 |
Fast Fourier transform (FFT) is one of the most useful tools and is widely used in signal processing. |
Fast Fourier transform (FFT) is one of the most useful tools and is widely used in signal processing, as shown in Figure 4. |
Figure 4. Fast Fourier Transform of a Cosine Summation Function resonating at 10, 20, 30, 40, and 50 Hz |
Figure 4. An example of Fast Fourier Transform |
Based on these assumptions, the study was conducted with a new labeling of the existing power data set, and then with the CNN-LSTM model we built, we found that both Train Loss and Test Loss were well developed with near zero convergence. |
Based on these assumptions, the study was conducted with a new labeling of the existing power data set, and then with the CNN-LSTM model we built, we found that both Train Loss and Test Loss were well developed with near zero convergence, as shown in Figure 14. |
8) general More examples are required to show how does the data analysis help in manufacturing management. The big question is how to evaluate that the artificial intelligent model is well trained and it is ready to be used in manifesting management. There should be an error indices / measures which show the accepted limits of error and the minimum allowable efficiency for the model training. I saw there is no such evaluation for the data analysis and prediction models. You may find some useful information in the following paper about the error indices https://doi.org/10.1016/j.jcde.2018.12.003
|
=> We appreciate this valuable comment. Thankfully, we also referred to the useful information in Muqdad Al-Juboori, et al.’s paper as the reviewer guided.
Parts Added (in red color) |
We have conducted supplementary experiments to confirm that both the preprocessing method and the applied model are superior to other deep learning-based models. The following table shows the performance results for each model. As can be seen in Table 2 (L425), the proposed model outperforms others in terms of error indices. |

Reviewer 2 Report
- The Introduction and related work can be reorganized together, and the authors should highlight the differences between the proposed algorithm and existing similar algorithms. Do not simply list all the algorithms studied from references, make the comparison in detail.
- Please check the number and name of the figures carefully.
- Page 7, line 257 and 258: It’s not only the techniques used for data preprocessing in this paper. As described in the subtitle, the most important part is the proposed algorithm used in this paper. The authors should revise the description of this section.
- Page 7, line 266 and 267, “It was observed that the temperature 266 change by time of the machine operation itself was larger than the change by temperature.” It’s hard to understand.
- The test platform should be described in detail.
- The author should explain in detail how the FFT is used for data transformation in this experiment.
- The author should explain the structure of the proposed CNN-LSTM algorithm in detail. What’s the function of different layers? Explain in detail the difference between the designed layer and classic CNN layer or LSTM layer.
- It’s better to know the range of data used for experiment without FFT transform since the author mentioned “15 sec average data at a sampling frequency of 100HZ” was used for experiment with FFT transform.
Author Response
We would greatly appreciate the referees’ valuable comments and suggestions. We hereby explain to clarify all the points brought up by the reviewers and the responses and are provided in blue color. We hope that the revised version and answers would help the understanding of the reviewers and the readers.
< Reviewer 2 Comment >
1) The Introduction and related work can be reorganized together, and the authors should highlight the differences between the proposed algorithm and existing similar algorithms. Do not simply list all the algorithms studied from references, make the comparison in detail.
|
=> We appreciate your valuable comment. We appended the following sentences, which provide the comparisons among all available algorithms, in the end of section 2(Related Works) according to your advice.
Parts Added (in red color) |
From the review of the preceding studies, we have observed that it is necessary to find an effective way in order to set the threshold of the abnormal pattern within the entire data were not labeled for fault detection task in manufacturing sites. For building the model which can find the abnormal pattern for fault detection, most of the preceding studies used the labeled data for the training. An algorithm most frequently used in previous studies is LSTM and the algorithms that is recently in research trials are PCA, SVM and VAN. The algorithms that has recently been newly tested to solve the data imbalance problem are SVM and VAN. In this paper, we tried to address these issues by the proposed Hybrid CNN-LSTM algorithm, which involves the Fast Fourier Transform (FFT) technique and Preprocessing the data set through FFT, enables us to detect anomalies that cannot be found in the existing time domain. |
2) Please check the number and name of the figures carefully. |
=> We appreciate your thankful comments. We checked the right location of the figure and reflected it as in the followings. In addition, we rephrased the right figure in the manuscript to correct the missed figure or misquoted figure.
We unified the figure caption over all the manuscript according to the journal requirements.
1) Font: Palatino Linoty 9, 2) Text alignment : Justify
For Figure 5 & 6, we modified the original caption in bold to one in plain.
Parts revised (in blue color) |
|
CNN (Convolutional Neural Networks) is a universally used algorithm for two-dimensional image classification and is structured by connecting the characteristic extraction neural network and the classification neural network in series. |
CNN (Convolutional Neural Networks) is a universally used algorithm for two-dimensional image classification and is structured by connecting the characteristic extraction neural network and the classification neural network in series, as shown in Figure 1. |
which uses internal circulation structure to reflect past learning into current learning through weight, as shown in Figure 1 [26]. |
which uses internal circulation structure to reflect past learning into current learning through weight, as shown in Figure 2 [26]. |
LSTM is one of the algorithms belonging to the RNN classification using the memory block shown in Figure 2 |
LSTM is one of the algorithms belonging to the RNN classification using the memory block shown in Figure 3 |
Fast Fourier transform (FFT) is one of the most useful tools and is widely used in signal processing. |
Fast Fourier transform (FFT) is one of the most useful tools and is widely used in signal processing, as shown in Figure 4. |
Figure 4. Fast Fourier Transform of a Cosine Summation Function resonating at 10, 20, 30, 40, and 50 Hz |
Figure 4. An example of Fast Fourier Transform |
Based on these assumptions, the study was conducted with a new labeling of the existing power data set, and then with the CNN-LSTM model we built, we found that both Train Loss and Test Loss were well developed with near zero convergence. |
Based on these assumptions, the study was conducted with a new labeling of the existing power data set, and then with the CNN-LSTM model we built, we found that both Train Loss and Test Loss were well developed with near zero convergence, as shown in Figure 14. |
3) Page 7, line 257 and 258: It’s not only the techniques used for data preprocessing in this paper. As described in the subtitle, the most important part is the proposed algorithm used in this paper. The authors should revise the description of this section.
This section offers a detailed discussion of the basic algorithms and techniques for data preprocessing used in this paper.
|
=> We appreciate your valuable suggestion. We modified the sentence according to your advice.
Parts revised (in blue color) |
|
This section offers a detailed discussion of the basic algorithms and techniques for data preprocessing used in this paper. |
This section offers a detailed discussion of the method proposed in this paper, where it includes data collection, the applied techniques for data preprocessing such as Fast Fourier Transform (FFT) and frequency domain analysis, and the algorithm proposed ultimately such as CNN-LSTM Algorithm. All of these are significant in that the abnormal pattern can be reliably detected through the integration of both ‘FFT-Frequency domain analysis’ and ‘CNN-LSTM’ which has not been tried yet. |
4) Page 7, line 266 and 267, “It was observed that the temperature 266 change by time of the machine operation itself was larger than the change by temperature.” It’s hard to understand.
It was observed that the temperature change by time of the machine operation itself was larger than the change by temperature. |
=> We appreciate this valuable comment. We modified the sentence and appended additional sentences to help the readers’ understanding much clear.
Parts revised (Blue-Colored) |
|
It was observed that the temperature change by time of the machine operation itself was larger than the change by temperature. |
The temperature measured from the sensor attached to the equipment continued increasing after the initial operating time, and it also displayed a pattern reaching to a peak at noon. It was observed that the change in temperature due to the generation of heat with the elapse of the operating time of the machine itself was much larger than the change in the external temperature, and we recognized that it made a considerably greater influence on the temperature measured from the sensor. |
5) The test platform should be described in detail. |
=> We appreciate this valuable comment. We appended the following paragraph, in the end of the description of ‘4. Proposed Method’ (L276), according to your advice.
Parts Added (in purple color) |
The test platform is explained in order as follows. First, we measured and collected three types of data, i.e. electric power, temperature and vibration data, in a second unit for 56 days, then performed preprocessing for appropriate labeling which correspond to normal operation of a machine or malfunctions due to external conditions. After taking FFT and conducting the relabeling, we apply to two representative algorithms of a typical ‘LSTM’ and a hybrid ‘CNN-LSTM’ (proposed) under the framework of ‘PyTorch’ and 3-GPUs-enabled deep-learning Server while flexibly adjusting hyper parameters for performance evaluation in accuracy and loss. |
6) The author should explain in detail how the FFT is used for data transformation in this experiment.
|
è We appreciate this valuable comment. We appended the following paragraph, which explains how FFT is used and what the overall process is like, in the end of Figure 5 according to your advice.
Parts Added (in purple color) |
The overall process including FFT is as shown in Figure 5. We average 15 time-periods data in a row with the labelling which we appropriately discriminated between normal and abnormal pattern in power data. Then, for well catching up and clearly visualizing the distinct time intervals for abnormal labelling, we take the FFT of the preceding time-domain averaged data. In our experiments, we need an interpolation in the frequency-domain, where feature observation in specific intervals can be spread into much wider frequency intervals under the assumption of Nyquist-Shannon sampling theorem. The Nyquist-Shannon Sampling Theorem implies that a bandlimited continuous-time signal can be sampled and perfectly reconstructed from its samples if the waveform is sampled over twice as fast as it is highest frequency component. For example, if we consider (0, 500Hz) frequency-domain signal, we need to take the sampling frequency of 1,000Hz or larger. Then, we extend the analysis to (0, 3,970Hz) for much more clear visualization in terms of interpolation, if necessary. From the interpolated frequency-domain data, the corresponding time-domain data are reconstructed and the relabeling is performed.
|
7) The author should explain the structure of the proposed CNN-LSTM algorithm in detail. What’s the function of different layers? Explain in detail the difference between the designed layer and classic CNN layer or LSTM layer.
|
=> We appreciate this valuable comment. We appended the following sentences in the end of section 4(Proposed Method) according to your advice.
Parts Added (in purple color) |
The CNN-LSTM consists of Convolutional layer, LSTM layer and Fully Connected layer. Each layer can adjust hyper parameters. Adjusting these parameters can affect performance depending on the characteristics of the training data. Our experiments measured optimal performance by adjusting parameters. The first Convolution layer where the kernel size is 5x1 and the number of filters is 20 received 15x1 data as input. It is difficult to extract stable features if the kernel size cannot cover a periodic cycle in time in data. Therefore, a 5x1 kernel was used to minimized information loss and extract local features. In the second convolution layer, the 50 filters were applied to the output of first layer to extract certain features. These extracted results help to better reflect the characteristics of data while reducing the number of parameters used in the following LSTM layer. Before applying it to the LSTM algorithm, the process of making two-dimensional data into one-dimensional data, the result of the convolution layer, is necessary. LSTM network had one hidden layer with 10 units. Finally, the output from LSTM layer entered the FCN layer consist of 10 units to classify normal and abnormal conditions. |
8) It’s better to know the range of data used for experiment without FFT transform since the author mentioned “15 sec average data at a sampling frequency of 100HZ” was used for experiment with FFT transform.
The average loss rate was lower when using CNN-LSTM algorithm after FFT transform using the 15 sec average data at a sampling frequency of 100 Hz than when using the CNN-LSTM algorithm without FFT transform in the training data set shown in Figure 12.
|
=> We appreciate this valuable comment. We appended the following paragraph in the end of Table 1 according to your advice.
Parts Added (in purple color) |
As mentioned before in Figure 4, FFT often helps for frequency component-characterized understanding by displaying the frequency components much clearer, even when the time-domain signal is difficult to recognize or discern periodic or cyclo-periodic features. The experiments without FFT are performed over the entire data set which is a second-based observed and measured for the last 56 days. Including specific intervals where power data abruptly increases, we evenly take the time-domain signal subsets and their corresponding labeling. Then, we apply to two representative algorithms of LSTM and hybrid CNN-LSTM, which are expected to provide good performances, while DNN does not reflect the memory characteristics of time series data and has considerable computational complexity as the number of layers increases and RNN has the disadvantage of long-term dependency problem, i.e. gradually decreasing weights as learning progresses. Therefore, we designed a novel idea that after extracting the frequency-domain characteristics for abnormal pattern via FFT and applying to the Nyquist-Shannon sampling theorem we could regroup a second-based time-domain data sets into 15 seconds-based ones while we observed abnormal patterns lasted for about 14 to 16 seconds. |

Reviewer 3 Report
There are lots of issues in this manuscript.
- This manuscript definitely needs to be improved by a native speaker or extensive English editing. There are some grammar issues that could be found throughout the manuscript.
- This manuscript needs to be reorganized. There are so many short paragraphs with a limited number of sentences and information. The authors should consider combining them.
- The introduction is shot with very limited information. Consider combining sections 1, 2, and 3 to improve the introduction.
- The methodology is simple and hard to follow. More figures and explanations are needed.
- All figures included in the results section need to be improved.
- The discussion needs to be separated from the conclusion. More discussion is needed by comparing the proposed studies with past studies to highlight your contributions. In addition to summarizing your work, the limitation and possible future research direction of this study needs to be included as well.
Author Response
We would greatly appreciate the referees’ valuable comments and suggestions. We hereby explain to clarify all the points brought up by the reviewers and the responses and are provided in blue color. We hope that the revised version and answers would help the understanding of the reviewers and the readers.
< Reviewer 3 Comment >
1) This manuscript definitely needs to be improved by a native speaker or extensive English editing. There are some grammar issues that could be found throughout the manuscript.
|
=> We appreciate your valuable comment. We tried to amend all the grammar errors and issues. This manuscript has been proofread by a style editor before the submission. If necessary, we will have it improved by native speaker once more.
Parts revised (in blue color) |
|
A summary statistics of the data used is shown in Table 1. |
The descriptive statistics for the data used is shown in Table 1. |
2) This manuscript needs to be reorganized. There are so many short paragraphs with a limited number of sentences and information. The authors should consider combining them.
|
=> We appreciate your thankful comments. We modified as followings according to your kind advice.
Parts revised (in blue color) |
|
The Deep Convolutional Neural Networks (DCNN) is currently showing high resolution for a variety of problems. However, time-series data recognition requires continuous label prediction rather than a single label, unlike normal data recognition. |
The Deep Convolutional Neural Networks (DCNN) is currently showing high resolution for a variety of problems, but time-series data recognition requires continuous label prediction rather than a single label, unlike normal data recognition. |
Rotating machinery appropriately operates under tough conditions and environments. As a result, it has a higher faulty rate compared with other parts in a mechanical system. |
Rotating machinery appropriately operates under tough conditions and environments and so it has a higher faulty rate compared with other parts in a mechanical system. |
Y. Sun proposed a cost-sensitive boosting algorithm introducing cost items into the learning framework of AdaBoost (Adaptive Boosting), a kind of Ensemble-based classifier. Y. Sun's study showed that the cost-sensitive boosting algorithm can improve predictive performance for minority classes [13]. |
Y. Sun proposed a cost-sensitive boosting algorithm introducing cost items into the learning framework of AdaBoost (Adaptive Boosting), a kind of Ensemble-based classifier and Y. Sun's study showed that the cost-sensitive boosting algorithm can improve predictive performance for minority classes [13]. |
Principal component analysis (PCA) is widely used in fault diagnosis. Z. Wang et al. proposed the data preprocessing method based on the Gap metric in the Riemann space to improve the performance of PCA in fault diagnosis. |
Principal component analysis (PCA) is widely used in fault diagnosis and Z. Wang et al. proposed the data preprocessing method based on the Gap metric in the Riemann space to improve the performance of PCA in fault diagnosis. |
3) The introduction is short with very limited information. Consider combining sections 1, 2, and 3 to improve the introduction.
|
=> We appreciate your valuable suggestion. We moved the following sentences in section 2 to introduction to improve short introduction with limited information.
Parts moved from Section 2(Related works) to Section1(Introduction) : in purple color |
According to the 2011 McKinsey Global Institute data, the manufacturing industry accumulates data notably larger than other industries. As of 2009, US companies have 966 petabytes of data for discrete manufacturing and 694 petabytes for process manufacturing industries such as chemical processing [2]. Equipment in the manufacturing industry is composed of automation equipment, and log data in numerical or text format is stored from the equipment or sensors installed nearby, and the operation of manufacturing facilities is managed based on these data. The operational management methodologies for the maintenance of manufacturing facilities include the â‘ reactive method, â‘¡ predictive method, â‘¢ proactive failure method, and â‘£ self-maintenance method [3]. Reactive maintenance refers to maintenance performed after a failure has occurred, and predictive maintenance is the method of performing maintenance to avoid failures according to conditions based on time. In addition, proactive maintenance is a method of eliminating the cause of the failure in advance, and self-maintenance is a method of self-maintaining from failure detection to action [3].
|
4) The methodology is simple and hard to follow. More figures and explanations are needed.
|
=> We appreciate this valuable comment. To help for the reader’s understanding in the methodology proposed, we replaced the original Figure 6 (Proposed CNN-LSTM algorithm, previously labelled as “Figure 5”) by a new one in more details. The supplementary descriptions include the altering process from time-series power data to vectorization (2D to 1D) through convolution layer, followed by LSTM and FC layer.
Parts Added |
The CNN-LSTM consists of Convolutional layer, LSTM layer and Fully Connected layer. Each layer can adjust hyper parameters. Adjusting these parameters can affect performance depending on the characteristics of the training data. Our experiments measured optimal performance by adjusting parameters. The first Convolution layer where the kernel size is 5x1 and the number of filters is 20 received 15x1 data as input. It is difficult to extract stable features if the kernel size cannot cover a periodic cycle in time in data. Therefore, a 5x1 kernel was used to minimized information loss and extract local features. In the second convolution layer, the 50 filters were applied to the output of first layer to extract certain features. These extracted results help to better reflect the characteristics of data while reducing the number of parameters used in the following LSTM layer. Before applying it to the LSTM algorithm, the process of making two-dimensional data into one-dimensional data, the result of the convolution layer, is necessary. LSTM network had one hidden layer with 10 units. Finally, the output from LSTM layer entered the FCN layer consist of 10 units to classify normal and abnormal conditions.
|
5) All figures included in the results section need to be improved. |
=> We appreciate this valuable comment. Including the structure of CNN-LSTM and all the figures (Figure 7 through Figure 15), we tried to enhance the resolution of all the figures, and adjust the labels, fonts, figure sizes, etc. for helping the much clearer understanding.
6) The discussion needs to be separated from the conclusion. More discussion is needed by comparing the proposed studies with past studies to highlight your contributions. In addition to summarizing your work, the limitation and possible future research direction of this study needs to be included as well.
|
=> We appreciate this valuable comment. According to your advice, we split "6. Discussion and conclusion" into “6. Discussion” and “7. Conclusion. We appended several sentences regarding the summary, the limitation and possible future research in the conclusion.
Parts moved to Conclusion: (in purple work) |
Based on the previous research Rui Yang conducted in 2018 [8], the data in the time-domain was converted to the frequency-domain through FFT, and a new abnormal pattern derivation process was performed. As a result of learning at various frequencies after conversion, it was confirmed that our model can easily find abnormal patterns and has excellent learning performance. By converting the data to the Frequency-Domain by FFT, it was possible to detect anomalies that cannot be found in the existing Time domain. It was verified through Loss and Accuracy that the learning performance in the proposed Hybrid algorithm, CNN-LSTM algorithm, is superior to that of widely used existing algorithms such as LSTM. This study was performed based on data generated during the manufacturing process over 3 months, and there were limitations in that the target period for detecting patterns was short. It cannot be ruled out that no abnormalities existed in the three-month period during which the set was recorded. If longer-term time series data were to be used for learning, it would be possible to establish a sophisticated model that can be applied to the industry immediately because it would be relatively more likely that abnormalities in the process have actually occurred. The data preprocessing method by FFT technique, which converts to the time-domain and frequency-domain, greatly helps to understand the abnormal pattern of unlabeled big data produced at the manufacturing site, and is judged to be an effective way to use deep learning methods with high learning performance without pre-labeling. The results of this study are judged to be available as a manufacturing failure prediction algorithm that alerts future failures and accidents in advance through past big data analysis in the manufacturing sector. |
Parts added (in red color) |
In this research work, our proposed Hybrid CNN-LSTM algorithm and the Fast Fourier Transform (FFT) technique present effective way to set threshold of the abnormal pattern from data were not labeled for fault detection task in manufacturing sites and showed that both train loss and test loss were well developed with near zero convergence with the lowest loss rate compared to existing models such as LSTM. The limitation of this study is that the research was performed for the data gathered during relatively short period. Our proposed model and method of preprocessing the data greatly helps to understand the abnormal pattern of unlabeled big data produced at the manufacturing site and can be a strong foundation for detecting the threshold of the abnormal pattern of big data occurring at manufacturing sites.
|

Reviewer 4 Report
The article has a potential, but needs improvement. Chapter "2. Related Work" is worth summarizing and presenting a concise conclusion at the end. My assessment lacks a concise introduction to Chapter "3. Background" – one sentence is not enough. In my opinion, you can’t go immediately straight to the technical description of the algorithms. I would consider splitting Chapter "6. Discussion and conclusion" – 6. Discussion; 7. Conclusion. Specific remarks:
- McKinsey Global Institute data (2011) and data from 2009 are already relatively old [lines 70–73]. That was a decade ago. And now? Is there more recent McKinsey Global Institute data?
- I got the impression (maybe it's subjective) that the "Related Work" chapter is somewhat chaotic; "Cracked," like a broken, and then glued vase. At some point the chapter ends ... and there is no summary or conclusions from the described research. It is worth summarizing the Related Work chapter. What generally resulted from all these studies?
- Line 151 – the Y. Sun surname is red – why?
- Line 201–202 – this has already been written in the introduction. I believe that starting chapters 3, 4 and 5 with only one short sentence is unnecessary. In particular, the sentence in the pre–chapter 5 is redundant.
- "5. Results – This section provides a concise and precise description of the results of our experiment and their interpretation." – it has already been written in the introduction. In addition, the chapter title speaks for itself.
- Short introductions to chapters should be written differently, maybe they should take the form of a synthesis of what is described in a given chapter?
- Line 387–388 – this sentence is superfluous. The title of the chapter speaks for itself.
- Line 414 – "Based on the previous research Yang and Rui conducted in 2018" – no reference to the source?
- Missing section: Author Contributions (?)

Author Response
We would greatly appreciate the referees’ valuable comments and suggestions. We hereby explain to clarify all the points brought up by the reviewers and the responses and are provided in blue color. We hope that the revised version and answers would help the understanding of the reviewers and the readers.
< Reviewer 4 Comment >
1) The article has a potential but needs improvement. Chapter "2. Related Work" is worth summarizing and presenting a concise conclusion at the end. My assessment lacks a concise introduction to Chapter "3. Background" – one sentence is not enough. In my opinion, you can’t go immediately straight to the technical description of the algorithms
2) I got the impression (maybe it's subjective) that the "Related Work" chapter is somewhat chaotic; "Cracked," like a broken, and then glued vase. At some point the chapter ends ... and there is no summary or conclusions from the described research. It is worth summarizing the Related Work chapter. What generally resulted from all these studies?
|
=> We appreciate your valuable comment. By appending several sentences in the beginning of “3. Background” and also inserting the summarizing paragraph in the end of “2. Related Works”, we tried to clarify the points the reviewer mentioned.
Parts Revised (in blue color) |
|
This section presents a detailed discussion of the basic algorithms and techniques used in this study. |
As we have investigated in related works of the previous section, there exist various application cases of typical machine learning algorithms. We hereby focus on the manufacturing big data analytics, in situation that a company’s business loss becomes too fatal because the repair cost of semiconductor manufacturing equipment’s breakdown is too huge and takes 3 to 6 months at least. This section presents a short introduction of the basic algorithms such as CNN (Convolutional Neural Networks), RNN and LSTM Algorithm, and techniques for data preprocessing such as Fast Fourier transform (FFT) to understand the modified algorithms and techniques used in this study. |
Parts Added (in red color) |
From the literature review above, we recognized that we often feel the necessity to find an effective way to set the threshold of the abnormal pattern in data which are not labeled for fault detection task in manufacturing sites. Most of previous studies utilized the labeled data for the training so that they built software learning models for fault detection pattern. The algorithm most frequently used in previous studies is LSTM and machine-learning algorithms that have been tried recently are PCA, SVM, and VAN. Out of those, SVM and VAN have recently been newly tested to solve the data imbalance problem. In this paper, we tried to address these issues by the proposed Hybrid CNN-LSTM algorithm and the Fast Fourier Transform (FFT) technique, where we perform preprocessing the data set through FFT which enables us to detect anomalies that cannot be found in the existing time domain. |
3) I would consider splitting Chapter "6. Discussion and conclusion" – 6. Discussion; 7. Conclusion. Specific remarks:
|
=> We appreciate this valuable comment. According to your advice, we split "6. Discussion and conclusion" into “6. Discussion” and “7. Conclusion. We appended several sentences regarding the summary, the limitation and possible future research in the conclusion.
Parts moved to Conclusion: in purple color |
Based on the previous research Rui Yang conducted in 2018 [8], the data in the time-domain was converted to the frequency-domain through FFT, and a new abnormal pattern derivation process was performed. As a result of learning at various frequencies after conversion, it was confirmed that our model can easily find abnormal patterns and has excellent learning performance. By converting the data to the Frequency-Domain by FFT, it was possible to detect anomalies that cannot be found in the existing Time domain. It was verified through Loss and Accuracy that the learning performance in the proposed Hybrid algorithm, CNN-LSTM algorithm, is superior to that of widely used existing algorithms such as LSTM. This study was performed based on data generated during the manufacturing process over 3 months, and there were limitations in that the target period for detecting patterns was short. It cannot be ruled out that no abnormalities existed in the three-month period during which the set was recorded. If longer-term time series data were to be used for learning, it would be possible to establish a sophisticated model that can be applied to the industry immediately because it would be relatively more likely that abnormalities in the process have actually occurred. The data preprocessing method by FFT technique, which converts to the time-domain and frequency-domain, greatly helps to understand the abnormal pattern of unlabeled big data produced at the manufacturing site, and is judged to be an effective way to use deep learning methods with high learning performance without pre-labeling. The results of this study are judged to be available as a manufacturing failure prediction algorithm that alerts future failures and accidents in advance through past big data analysis in the manufacturing sector. |
Parts added (in red color) |
In this research work, our proposed Hybrid CNN-LSTM algorithm and the Fast Fourier Transform (FFT) technique present effective way to set threshold of the abnormal pattern from data were not labeled for fault detection task in manufacturing sites and showed that both train loss and test loss were well developed with near zero convergence with the lowest loss rate compared to existing models such as LSTM. The limitation of this study is that the research was performed for the data gathered during relatively short period. Our proposed model and method of preprocessing the data greatly helps to understand the abnormal pattern of unlabeled big data produced at the manufacturing site and can be a strong foundation for detecting the threshold of the abnormal pattern of big data occurring at manufacturing sites.
|
4) McKinsey Global Institute data (2011) and data from 2009 are already relatively old [lines 70–73]. That was a decade ago. And now? Is there more recent McKinsey Global Institute data?.
According to the 2011 McKinsey Global Institute data, the manufacturing industry accumulates data notably larger than other industries. As of 2009, US companies have 966 petabytes of data for discrete manufacturing and 694 petabytes for process manufacturing industries such as chemical processing [2].
|
=> We appreciate your valuable suggestion. We tried to find recent statistics data regarding big data generation or accumulation amount in manufacturing industry because we understood your point. However, we did not find more recent statistics data yet. Book named "Advanced Intelligent Systems for Sustainable Development" says the amount of big data generated every day is over 2.5 Quintillion bytes. We do not think it is easy to provide recent statistics data regarding big data generation or accumulation amount in manufacturing industry, because the amount of big data is tremendous.
5) Line 151 – the Y. Sun surname is red – why?
Y. Sun's study showed that the cost-sensitive boosting algorithm can improve predictive performance for minority classes [13]. |
=> We appreciate your kind comment. It is a simple mistake during correcting the spelling. We modified the surname into black color.
6) Line 201–202 – this has already been written in the introduction. I believe that starting chapters 3, 4 and 5 with only one short sentence is unnecessary. In particular, the sentence in the pre–chapter 5 is redundant.
This section presents a detailed discussion of the basic algorithms and techniques used in this study. [Section 3]
This section offers a detailed discussion of the basic algorithms and techniques for data preprocessing used in this paper. [Section 4]
This section provides a concise and precise description of the results of our experiment and their interpretation. [Section 5] |
=> We appreciate this valuable comment. We modified as follows according to your advice.
Parts Revised (in blue color) |
|
This section presents a detailed discussion of the basic algorithms and techniques used in this study. |
This section presents a short introduction of the basic algorithms such as CNN (Convolutional Neural Networks), RNN and LSTM Algorithm and techniques for data preprocessing such as Fast Fourier transform (FFT) to understand the modified algorithms and techniques used in this study. |
This section offers a detailed discussion of the basic algorithms and techniques for data preprocessing used in this paper. |
This section offers a detailed discussion of the method proposed in this paper including data collection, techniques for data preprocessing such as Fast Fourier Transform (FFT) and the algorithms used such as CNN-LSTM Algorithm. |
This section provides a concise and precise description of the results of our experiment and their interpretation. |
We tried to train the dataset by LSTM Algorithm at first, but the learning was not achieved because there was no tendency to decrease loss. And then we tried to train the dataset by our proposed CNN-LSTM Algorithm, however there was still loss showing a lack of convergence even though the accuracy of the learning was enhanced. To solve a lack of convergence in loss, we converted the data set from the time domain data to the frequency domain using FFT transform technique. To find patterns that better detect abnormal patterns in the data set, we performed FFT transform on various frequencies using the 15 sec average data. We can get the best result with the lowest loss rate with our proposed CNN-LSTM Algorithm after FFT transform at 1,000 Hz. |
7) "5. Results – This section provides a concise and precise description of the results of our experiment and their interpretation." – it has already been written in the introduction. In addition, the chapter title speaks for itself
8) Short introductions to chapters should be written differently, maybe they should take the form of a synthesis of what is described in a given chapter?.
|
=> We appreciate this valuable comment. Reflecting the reviewer’s opinion that the sentence has already been written in the introduction, we modified the introduction of the section “5. Results” as follows.
Parts Revised (in blue color) |
|
This section provides a concise and precise description of the results of our experiment and their interpretation. |
We tried to train the preprocessed dataset by LSTM Algorithm at first, but the learning was not achieved appropriately because there was no tendency to decrease loss. After that, we moved towards training the dataset by our proposed CNN-LSTM Algorithm, but a lack of convergence in loss performance was observed even though the accuracy of the learning was improved highly. To cope with a lack of convergence in loss, we converted the data set from the time domain to the frequency domain using FFT technique. Then, to find frequency-domain segments that better detect abnormal patterns in the data set, we performed IFFT on various frequencies using the 15 seconds averaged data. We could ultimately get the best results with the lowest loss rate with our proposed CNN-LSTM Algorithm after taking FFT at 1,000 Hz. |
9) Line 387–388 – this sentence is superfluous. The title of the chapter speaks for itself. . This section provides a concise discussion of the experiment and the conclusions that can be drawn. |
=> We appreciate this valuable comment. We deleted the sentence according to your kind advice.
10) Line 414 – "Based on the previous research Yang and Rui conducted in 2018" – no reference to the source?
Based on the previous research Yang and Rui conducted in 2018, the data in the time-domain was converted to the frequency-domain through FFT, and a new abnormal pattern derivation process was performed. |
=> We appreciate this valuable comment. We appended the reference and modified the spelling errors as follows.
Parts Revised (in blue color) |
|
Yang and Rui |
Rui Yang |
Based on the previous research Yang and Rui conducted in 2018, |
Based on the previous research Rui Yang conducted in 2018 [8], |
11) Missing section: Author Contributions (?) |
=> We appreciate this valuable comment. We appended “Author contributions” and “Funding” just before the part of “Conflicts of Interest”.
Parts Added (in purple color) |
Author Contributions: Conceptualization, Jeong-Hee Lee and Tae-Eung Sung; Data curation, Jongseok Kang and We Shim; Formal analysis, Jongseok Kang and We Shim; Methodology, Jeong-Hee Lee, Hyun-Sang Chung and Tae-Eung Sung; Writing – original draft, Jeong-Hee Lee; Writing – review & editing, Tae-Eung Sung. Funding: This research received no external funding. |

Round 2
Reviewer 3 Report
I appreciate the efforts made by the authors for improving the manuscript. I think the manuscript has been improved significantly, it should be accepted for publication after minor revisions. Please find my comments below.
- Introduction. There are several small paragraphs that should be considered to be merged together.
- Related work. This section needs to be re-organized. It can be grouped into several categories, then followed by a literature review in each group. it's not necessary to separate each study review by paragraphs.
- Backgrounds and proposed methods. Good job here.
- Results. I do like all the plots included in this section to support your research results. However, the text on the plots are so small, please make it readable.
- Discussion. very limited discussion, it needs to be expanded with more literature comparison to show the contributions you made here.
- Conclusions. In addition to summarize your research results, limitations of this study and potential future research direction should also be included in this section.
Author Response
We would greatly appreciate the referees’ valuable comments and suggestions. We hereby explain to clarify all the points brought up by the reviewers and the responses are provided in blue color. We hope that the revised version and the answers to the comments would help the understanding of both reviewers and readers.
< Reviewer 3 Comment >
appreciate the efforts made by the authors for improving the manuscript. I think the manuscript has been improved significantly, it should be accepted for publication after minor revisions. Please find my comments below.
1) .Introduction. There are several small paragraphs that should be considered to be merged together.
|
è We appreciate your valuable comment. We merged several small paragraphs together according to your kind advice.
Parts revised (in blue color) |
|
Recently, the research of big data in scientific and technological utilization has been rapidly increasing. There is an increasing trend of attempts to improve the competence and competitiveness of companies and discover new business items using big data information in manufacturing industry. |
Recently, the research of big data in scientific and technological utilization has been rapidly increasing and there is an increasing trend of attempts to improve the competence and competitiveness of companies and discover new business items using manufacturing big data information in manufacturing industry. |
Data obtained through sensors at the manufacturing site plays an important role in the successful operation of the manufacturing plant. Manufacturing company can get the overall quality improvement and cost reduction through defect tracking and predictive function improvement using machine learning. |
Data obtained through sensors at the manufacturing site plays an important role in the successful operation of the manufacturing plant and manufacturing company can get the overall quality improvement and cost reduction through defect tracking and predictive function improvement using machine learning. |
According to the 2011 McKinsey Global Institute data, the manufacturing industry accumulates data notably larger than other industries. As of 2009, US companies have 966 petabytes of data for discrete manufacturing and 694 petabytes for process manufacturing industries such as chemical processing [2]. |
According to the 2011 McKinsey Global Institute data, the manufacturing industry accumulates data notably larger than other industries and as of 2009, US companies have 966 petabytes of data for discrete manufacturing and 694 petabytes for process manufacturing industries such as chemical processing [2]. |
In manufacturing sites, a 24-hour continuous, seamless process is in progress, and sometimes situations occur in which the process environment changes unexpectedly. The manufacturing process repeats the production line stops and resumptions due to equipment maintenance or other reasons, so in the database there may accumulate data that exceeds the normal range or data that results in significant deviation due to malfunctioning sensors though in normal range. |
In manufacturing sites, a 24-hour continuous, seamless process is in progress, and sometimes situations occur in which the process environment changes unexpectedly and the manufacturing process repeats the production line stops and resumptions due to equipment maintenance or other reasons, so in the database there may accumulate data that exceeds the normal range or data that results in significant deviation due to malfunctioning sensors though in normal range. |
Labelling of anomalous data is exceedingly difficult without domain knowledge of the manufacturing process. Collaboration with manufacturing process experts enables data analysis for deep learning after labeling abnormal data patterns, but it is practically difficult and time-consuming for data analysis professionals to obtain help from manufacturing process experts at actual manufacturing sites. |
Labelling of anomalous data is exceedingly difficult without domain knowledge of the manufacturing process and collaboration with manufacturing process experts enables data analysis for deep learning after labeling abnormal data patterns, but it is practically difficult and time-consuming for data analysis professionals to obtain help from manufacturing process experts at actual manufacturing sites. |
The second section reviews current studies of big data-based machine learning. The third section describes the basic algorithms and techniques used in this study in detail. |
The second section reviews current studies of big data-based machine learning and the third section describes the basic algorithms and techniques used in this study in detail. |
The fourth section explains the method proposed in our study. Lastly, the fifth section contains the result of this study, followed by the final section presenting the discussion and conclusion. |
The fourth section explains the method proposed in our study and lastly, the fifth section contains the result of this study, followed by the final section presenting the discussion and conclusion. |
2) Related work. This section needs to be re-organized. It can be grouped into several categories, then followed by a literature review in each group. it's not necessary to separate each study review by paragraphs.
|
è We appreciate your thankful comments. We appended the subsections of 2.1 and 2.2 as follows and re-organized the literature reviews into each subsection group. We colored the sentences that its location is changed in purple. In addition, we renumbered the references in the body, and we marked in blue color in [L138] through [L208] when the location of the literature reviews is changed.
Parts added (in red color) |
. 2.1. Application of machine learning algorithms for the detection of fault and abnormal patterns 2.2. Application of machine learning algorithms for solving the data imbalance problem
|
3) Backgrounds and proposed methods. Good job here. |
è We appreciate your kind comment. Thank you so much for your great help!
4) .Results. I do like all the plots included in this section to support your research results. However, the text on the plots are so small, please make it readable.
|
è We appreciate this valuable comment. To reflect your kind advices, we have revised some of the pictures and plots that are less readable. Especially, we made the font size and the overall figure size much bigger. We indicated the changed figures in red color.
Parts revised |
The pictures and plots that we modified is as follows. Figure 3, Figure 4, Figure 7, Figure 8, Figure 9, Figure 10, Figure 11, Figure 12, Figure 13, Figure 14, Figure 15. |
5) Discussion. very limited discussion, it needs to be expanded with more literature comparison to show the contributions you made here.
|
è We appreciate this valuable comment. We appended a volume of sentences to show our contributions we made and mentioned more literature comparison as below.
Parts Added (in red color) |
Most organizations use enterprise software, which makes use of rule-based processing to automate business processes including manufacturing process and this task-based automation has helped organizations in improving their productivity in a few specific processes, but this rule-based software cannot self-learn and improve with experience. By analyzing the data, the AI systems can draw conclusions on the machine’s condition and detect irregularities to provide predictive maintenance. A strongly coupled digitized system in the manufacturing industry improves the overall quality and reduces the cost by improving defect tracking and forecasting abilities [34]. Manufacturing systems engineering will be greatly affected by advances in technology, including cheaper, ubiquitous sensors, increasing computational speeds, the ability to hold more data and move it faster, and artificial intelligence in the future [35]. We expect our proposed pattern detection model using a deep learning algorithm could be an effective way for the decision-making to determine proper timing of predictive maintenance with enhanced pattern tracking ability. |
References Added |
like 34. Markets and Markets, Artificial Intelligence in Manufacturing Market-Global forecast to 2025, December 2018. 35. 35. Gershwin, Stanley B. "The future of manufacturing systems engineering." International Journal of Production Research 56.1-2 (2018): 224-237. 35. |
6) Conclusions. In addition to summarize your research results, limitations of this study and potential future research direction should also be included in this section. |
è We appreciate this valuable comment. We appended the limitations of the study and our future research direction as follows, furtherly associating with the possibility of applying to the predictive maintenance (PdM) fields.
Parts revised (in blue color) |
|
The limitation of this study is that the research was performed for the data gathered during relatively short period.
|
The limitations of this study are that the study was performed on the data collected in a relatively short period of time and that only power data out of various types of data (e.g. vibration, temperature, humidity data, etc.) were analyzed for characterizing the abnormal conditions of an equipment or a machinery. In future studies, we will move towards conducting an extensive study that increases the reliability of the current research by analyzing the pattern detection model we propose over long-term time series data for a year or longer and all available data including temperature and pressure data besides power data will be applied to analyze for detecting anomalous patterns. |
The results of this study are judged to be available as a manufacturing failure prediction algorithm that alerts future failures and accidents in advance through past big data analysis in the manufacturing sector. |
The results of this study are judged to be available as a manufacturing failure prediction algorithm that alerts future failures and accidents in advance through past big data analysis in the manufacturing sector. In case of semiconductor manufacturing and processing equipment, it takes huge amount of fixing cost and over 2 or 3 months’ repair time when it is out of order. Recently, the fields of predictive maintenance (PdM) have been standing out due to its significance in order to prevent unexpected stops of a machinery in the manufacturing industry. Therefore, we anticipate that our extensive study over long-term manufacturing-related data in various types would contribute to or develop the PdM which monitors the performance and condition of equipment during normal operation to reduce the likelihood of failures. |
